

Atmospheric
Measurement
Techniques

# A new TROPOMI product for tropospheric NO$_2$ columns over East Asia with explicit aerosol corrections

**Mengyao Liu**[1,a], **Jintai Lin**[1], **Hao Kong**[1], **K. Folkert Boersma**[2,3], **Henk Eskes**[2], **Yugo Kanaya**[4], **Qin He**[5], **Xin Tian**[6,7], **Kai Qin**[5], **Pinhua Xie**[6,7,8], **Robert Spurr**[9], **Ruijing Ni**[1], **Yingying Yan**[10], **Hongjian Weng**[1], and **Jingxu Wang**[1]

[1]Laboratory for Climate and Ocean-Atmosphere Studies, Department of Atmospheric and Oceanic Sciences,
School of Physics, Peking University, Beijing, China
[2]R&D Satellite Observations Department, Royal Netherlands Meteorological Institute, De Bilt, the Netherlands
[3]Meteorology and Air Quality department, Wageningen University, Wageningen, the Netherlands
[4]Research Institute for Global Change, Japan Agency for Marine-Earth Science and Technology (JAMSTEC),
Yokohama 2360001, Japan
[5]School of Environment and Geoinformatics, China University of Mining and Technology,
 Xuzhou, Jiangsu, 221116, China
[6]Institutes of Physical Science and Information Technology, Anhui University, Hefei, 230601, China
[7]Key Laboratory of Environmental Optics and Technology, Anhui Institute of Optics and Fine Mechanics,
Chinese Academy of Science, Hefei, 230031, China
[8]CAS Center for Excellence in Urban Atmospheric Environment, Institute of Urban Environment,
Chinese Academy of Sciences, Xiamen 361021, China
[9]RT Solutions Inc., Cambridge, Massachusetts 02138, USA
[10]Department of Atmospheric Sciences, School of Environmental Studies,
China University of Geosciences (Wuhan), Wuhan, China
[a]now at: R&D Satellite Observations Department, Royal Netherlands Meteorological Institute, De Bilt, the Netherlands

**Correspondence:** Jintai Lin (linjt@pku.edu.cn)

**Abstract.** We present a new product with explicit aerosol corrections, POMINO-TROPOMI, for tropospheric nitrogen dioxide (NO$_2$) vertical column densities (VCDs) over East Asia, based on the newly launched TROPOspheric Monitoring Instrument with an unprecedented high horizontal resolution. Compared to the official TM5-MP-DOMINO (OFFLINE) product, POMINO-TROPOMI shows stronger concentration gradients near emission source locations and better agrees with MAX-DOAS measurements ($R^2 = 0.75$; NMB = 0.8 % versus $R^2 = 0.68$, NMB = −41.9 %). Sensitivity tests suggest that implicit aerosol corrections, as in TM5-MP-DOMINO, lead to underestimations of NO$_2$ columns by about 25 % over the polluted northern East China region. Reducing the horizontal resolution of a priori NO$_2$ profiles would underestimate the retrieved NO$_2$ columns over isolated city clusters in western China by 35 % but with overestimates of more than 50 % over many offshore coastal areas. The effect of a priori NO$_2$ profiles is more important under calm conditions.

## 1 Introduction

Nitrogen oxides (NO$_x$ = NO + NO$_2$) are crucial gaseous pollutants in the troposphere. NO$_x$ leads to the production of particulate matter and ozone (O$_3$) and enhances levels of oxidants in the troposphere (Shindell et al., 2009), which affect air quality (Dentener et al., 2003) and human health (Hoek et al., 2013). Satellite remote sensing is widely used to monitor levels of nitrogen dioxide (NO$_2$) pollution worldwide (McLinden et al., 2014; Krotkov et al., 2016; Ott et al., 2010; Russell et al., 2011; Lin et al., 2010). The TRO-

POspheric Monitoring Instrument (TROPOMI), which was jointly developed by the Netherlands and Europe Space Agency (ESA) (Veefkind et al., 2012) and was launched on 13 October 2017, is a UV–visible (Vis)–near-infrared–shortwave infrared backscattering sensor on board the sun-synchronous Sentinel-5 Precursor (S5P) satellite with an overpass time of 13:30 local solar time. With a wide swath of 2600 km and an unprecedentedly high horizontal resolution of 3.5 km × 7 km, (3.5 km × 5.5 km since 6 August 2019) TROPOMI achieves daily global coverage. This high horizontal resolution and good spatial coverage, combined with the very high signal-to-noise ratio, enables the instrument to resolve NO₂ pollution from point sources, medium-size urban centers, highways, or rivers, tasks that were difficult to achieve before.

Retrievals of tropospheric NO₂ vertical column densities (VCDs) in the UV–Vis spectral range from satellite instruments consist of three steps: (1) using differential optical absorption spectroscopy (DOAS) to fit the slant column density (SCD) of NO₂ along the light path, (2) subtracting the stratospheric contribution from the SCD to obtain the tropospheric SCD, and (3) converting the tropospheric SCD to the tropospheric VCD by using the calculated air mass factor (AMF). For TROPOMI, the random uncertainty in the total SCDs is $\sim 0.6 \times 10^{15}$ molec. cm$^{-2}$, considerably smaller than for the Ozone Monitoring Instrument (OMI; $\sim 0.8 \times 10^{15}$ molec. cm$^{-2}$; Zara et al., 2018). The (total or stratospheric) bias is generally between 0 % and −10 % according to SAOZ observations (Eskes et al., 2019), which meets the error requirements as defined in the S5P Calibration and Validation Plan (Goryl et al., 2017). The calculation of the AMF introduces the dominant source of error in the retrieved tropospheric NO₂ VCDs over polluted areas (Boersma et al., 2004, 2011, 2018[TS2]; Lorente et al., 2017; Lin et al., 2014). The median negative biases of the daily comparisons between tropospheric VCDs from the Dutch official TM5-MP-DOMINO (OFFLINE) product and MAX-DOAS measurements are generally less than 50 % (the allowable bias is 25 %–50 %; Goryl et al., 2017) but quite variable with the stations and NO₂ levels, especially at polluted locations (Eskes et al., 2018; van Geffen et al., 2019).

TM5-MP-DOMINO uses implicit aerosol corrections by assuming aerosols to be "effective clouds", as assumed in most satellite NO₂ products except POMINO (Liu et al., 2019; Lin et al., 2014, 2015). In addition, TM5-MP-DOMINO employs a priori NO₂ profiles from the TM5 model at a relatively coarse horizontal resolution (1° × 1°; Williams et al., 2017). The implicit aerosol corrections (Lin et al., 2014; Lorente et al., 2017; Liu et al., 2019) and the coarse horizontal resolution of a priori NO₂ profile data (Laughner et al., 2016; Russell et al., 2011; McLinden et al., 2014) may be the largest sources of the large biases observed between TM5-MP-DOMINO and MAX-DOAS. Based on previous studies for OMI, implicit aerosol corrections can lead to uncertainties of more than 50 % over polluted ar-

eas with high aerosol loadings like China (Lin et al., 2014; Lorente et al., 2017; Liu et al., 2019). Eskes et al. (2018) showed that using a priori NO₂ profiles from the regional CAMS model at a 12 km × 12 km resolution to replace the TM5 NO₂ profiles increases the retrieved NO₂ VCDs by $\sim 0$ % to 50 % over western Europe depending on the location.

Here we present a new TROPOMI tropospheric NO₂ VCD product over East Asia, namely POMINO-TROPOMI. This product is based on our POMINO algorithm (Liu et al., 2019; Lin et al., 2014, 2015) previously applied to OMI. POMINO-TROPOMI improves upon TM5-MP-DOMINO by employing explicit aerosol corrections and using high-resolution ($\sim 25$ km) a priori NO₂ profiles, among other improvements. POMINO-TROPOMI NO₂ VCD data over July–October 2018 are presented and validated by MAX-DOAS measurements, along with additional sensitivity tests of the effects of aerosol representations and a priori NO₂ profiles.

## 2 Method and data

### 2.1 POMINO-TROPOMI retrieval algorithm and product

As one of the UV–Vis backscatter instruments to observe NO₂, TROPOMI inherits much of the design of OMI (Veefkind et al., 2012). Thus the POMINO-TROPOMI algorithm here largely follows our previous POMINO algorithm (Liu et al., 2019), with some modifications to adapt to its high horizontal resolution and different cloud retrieval procedure.

The POMINO-TROPOMI algorithm focuses on improving the calculation of tropospheric AMF. It thus takes the tropospheric SCD data from TM5-MP-DOMINO (OFFLINE), which are obtained by fitting the 405–465 nm wavelength range with the DOAS method. Our tropospheric AMF calculation is done for 437.5 nm, following TM5-MP-DOMINO.

We use the parallelized linearized discrete ordinate radiative transfer (LIDORT)-driven AMFv6 package to derive tropospheric AMFs via online pixel-specific radiative transfer calculations, with no use of lookup tables. Our algorithm explicitly accounts for aerosol optical effects and anisotropic properties of surface reflectance, uses daily a priori aerosol and NO₂ profiles from the simulation of nested GEOS-Chem model (0.25° lat × 0.3125° long; Zhang et al., 2016), and further uses aerosol optical depth (AOD) data from Moderate Resolution Imaging Spectroradiometer (MODIS/Aqua) to correct GEOS-Chem simulated aerosols on a monthly basis. Figure 1 shows the procedure of using the AMFv6 package to derive the tropospheric NO₂ VCDs of POMINO-TROPOMI. Table S1 in the Supplement shows the retrieval parameters in POMINO-TROPOMI, in comparison with those in TM5-MP-DOMINO and POMINO v2.

The independent pixel approximation (IPA) is used to calculate AMF as a linear combination of a cloudy AMF ($M_{cld}$)

Atmos. Meas. Tech., 13, 1–13, 2020
https://doi.org/10.5194/amt-13-1-2020

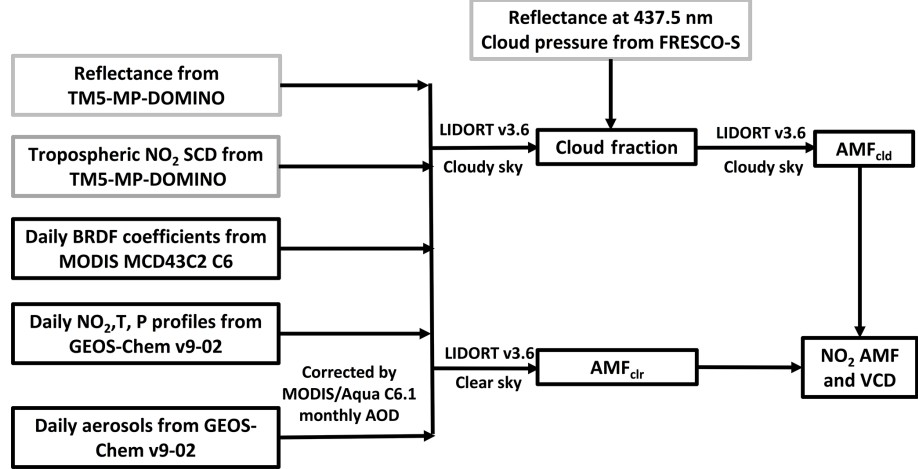

**Figure 1.** Flowchart of the POMINO-TROPOMI algorithm. The grey rectangles represent the parameters from the TM5-MP-DOMINO (OFFLINE) product.

and a clear-sky AMF ($M_{clr}$) as follows:

$$M = wM_{cld} + (1 - w)M_{clr}. \tag{1}$$

$w$ is the cloud radiation fraction (CRF) calculated by

$$w = \frac{f_{eff}I_{cld}}{R} = \frac{f_{eff}I_{cld}}{f_{eff}I_{cld} + (1 - f_{eff})I_{clr}}, \tag{2}$$

where $I_{cld}$ denotes the radiance from the cloudy part of the pixel, $I_{clr}$ the radiance from the clear-sky part of the pixel, $f_{eff}$ the cloud fraction (CF), and $R$ the total scene radiance. Retrieval of cloud properties is a prerequisite for NO₂ retrieval. We take the effective cloud pressure (CP) from the FRESCO-S algorithm (van Geffen et al., 2019), which uses the O₂ A band (around 758 nm) for TROPOMI trace gas retrievals. We recalculate $w$ and $f_{eff}$ at the NO₂ fitting wavelength (437.5 nm). The online CF calculation is similar to that for TM5-MP-DOMINO (Arnoud et al., 2017; van Geffen et al., 2019) but with an explicit aerosol correction to be consistent with the following NO₂ retrieval and to remove the aerosol signal from the retrieved CF data.

For explicit aerosol corrections in this study, we take daily aerosol simulation results from the GEOS-Chem v9-02 nested model over East Asia, followed by a monthly AOD correction using MODIS/Aqua C6.1 AOD data. Our future study will use the Cloud-Aerosol Lidar with Orthogonal Polarization (CALIOP) aerosol extinction vertical profiles to further improve the modeled aerosol profiles. Figure 2b shows the spatial distribution of AOD in July 2018 used in clouds and NO₂ retrievals. The AOD distribution is consistent ($R = 0.42$; $N = 1447$) with that of near-surface PM₂.₅ mass concentration measurements (Fig. 2a) taken from the Ministry of Ecology and Environment of China (MEE); the difference between AOD and near-surface PM₂.₅ is expected because they represent different parameters of aerosols.

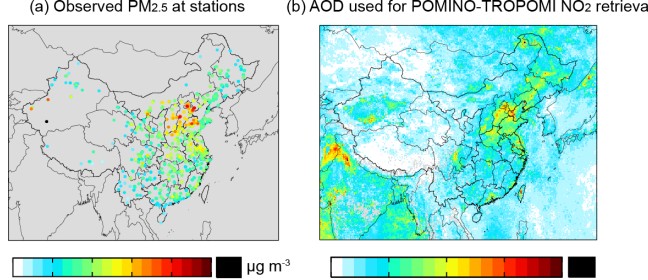

**Figure 2. (a)** Observed near-surface PM₂.₅ mass concentrations averaged over July 2018. Results are sampled at the times of valid TROPOMI data. **(b)** AOD data on a 0.05° × 0.05° grid used for the retrieval of POMINO-TROPOMI NO₂ VCDs in July 2018.

The criteria to select valid pixels in this study are as follows. We exclude pixels with viewing zenith angles (VZAs) greater than 80°, with high albedos caused by ice or snow on the ground, or with a quality flag (from TM5-MP-DOMINO) of less than 0.5. To screen out cloudy scenes, we discard pixels with CRFs greater than 50 % in the POMINO-TROPOMI product.

In addition to our formal POMINO-TROPOMI product (referred to as Case REF), we use sensitivity cases to evaluate the impacts of aerosol corrections (explicit versus implicit) and the horizontal resolution of a priori NO₂ profiles (Cases 1 and 2 in Table 2). Two additional cases (Cases 3 and 4) concern the treatment of CP in combination with the choice of aerosols and surface reflectance. Specifically, using the CP data directly from FRESCO-S means that our retrieval algorithm does not perfectly account for the effect of aerosols on clouds. Our retrievals consider the anisotropy of the surface via (bidirectional reflectance distribution function) BRDF effects, while a Lambertian surface is used in

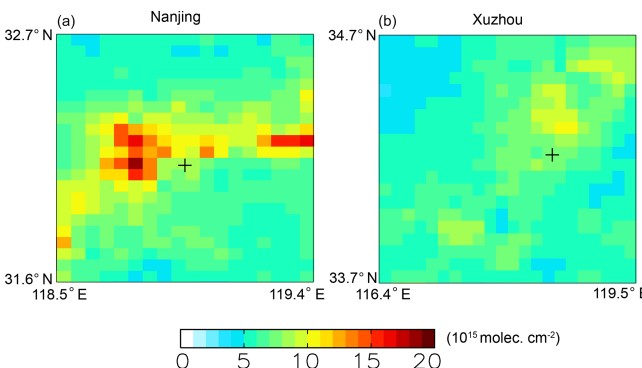

**Figure 3.** Spatial distributions of POMINO-TROPOMI NO$_2$ VCDs (on a $0.05° \times 0.05°$ grid) around **(a)** Nanjing and **(b)** Xuzhou MAX-DOAS measurement sites in July 2018. The MAX-DOAS sites are marked as "+".

deriving the FRESCO-S CP. To ensure sampling consistency, the pixels used in all cases are selected based on the CRF values in Case REF.

## 2.2 Ground-based MAX-DOAS measurements

We use ground-based MAX-DOAS measurements to validate the POMINO-TROPOMI NO$_2$ product. The MAX-DOAS measurements are from two suburban stations (Xuzhou and Nanjing) and one remote station (Fukue; Kanaya et al., 2014). Table 1 shows the geographical and time information of each MAX-DOAS site, and Sect. S1 in the Supplement describes each instrument in detail. Although Xuzhou and Nanjing are both classified as suburban sites located at university campuses, the NO$_2$ spatial distributions around the two sites are very different. The spatial distribution of NO$_2$ VCDs is relatively smooth around Xuzhou, whereas the VCDs exhibit a strong spatial gradient around Nanjing (Fig. 3).

A consistent spatiotemporal sampling is crucial in comparing satellite measurements and MAX-DOAS data (Boersma et al., 2018; Lin et al., 2014; Liu et al., 2019; Wang et al., 2017). We average all valid MAX-DOAS data within $\pm 1$ h of the TROPOMI overpass time to obtain daily values for comparison. To reduce the influence of local events, we exclude all MAX-DOAS data whose standard deviations within the 2 h exceed 20 % of their mean values. We average all valid pixels within 5 km of each MAX-DOAS site to represent the respective daily satellite data. Section S2 shows how the validation results are affected by the sampling choice.

## 3 Results

### 3.1 POMINO-TROPOMI NO$_2$ VCDs over East Asia

Figure 4b shows the spatial distribution of POMINO-TROPOMI tropospheric NO$_2$ VCDs over East Asia on

a $0.05° \times 0.05°$ grid in July 2018. High VCD values ($> 3 \times 10^{15}$ molec. cm$^{-2}$) are shown over polluted areas such as East China and India, and low values ($< 1 \times 10^{15}$ molec. cm$^{-2}$) are shown over the open ocean and the Tibetan Plateau. For comparison, the colored dots in Fig. 4a visualize the near-surface NO$_2$ concentrations observed at the MEE sites at the overpass time of TROPOMI. In both the VCD and the near-surface concentration maps (Fig. 4a and b), hotspots like urban centers and isolated sources can be seen clearly, due to the short lifetimes of NO$_x$ in summer. The spatial correlation is about 0.55 ($N = 1458$) between the VCD and the near-surface concentration distributions.

Figure 4c shows the spatial distribution of TM5-MP-DOMINO (OFFLINE) NO$_2$ VCDs for comparison. The general distribution of TM5-MP-DOMINO is consistent with that of POMINO-TROPOMI with a correlation coefficient of 0.97 ($N = 1\,091\,154$). However, TM5-MP-DOMINO NO$_2$ VCD values are lower than POMINO-TROPOMI by about 35 % averaged over the whole domain (Fig. 4d), by $-37$ %–68 % over cleaner areas (POMINO-TROPOMI $< 5 \times 10^{15}$ molec. cm$^{-2}$), and by 0 %–66 % over more polluted areas (POMINO-TROPOMI $\geq 5 \times 10^{15}$ molec. cm$^{-2}$). TM5-MP-DOMINO does not show strong local signals at pollution hotspots like the urban center of Beijing (Fig. 4c). Over these hotspot locations, TM5-MP-DOMINO is lower than POMINO-TROPOMI by up to $5 \times 10^{15}$ molec. cm$^{-2}$ (Fig. 4e).

Figure 5a and b present the two products over Beijing and surrounding areas, showing a much weaker spatial gradient of NO$_2$ VCDs from Beijing urban center to its outskirts in TM5-MP-DOMINO than in POMINO-TROPOMI. The corresponding histograms and Gaussian fittings in Fig. 5c also show a lower mean value and a smaller standard deviation of TM5-MP-DOMINO than POMINO-TROPOMI. These results highlight the important differences between the two products at fine scales.

We further compare the two satellite products with ground-based MAX-DOAS measurements at three sites. The scatterplots in Fig. 6a and b compare the NO$_2$ VCDs on 63 d (from 109 pixels) over July–October 2018 with their MAX-DOAS counterparts. Different colors differentiate the sites. POMINO-TROPOMI captures the day-to-day variability in MAX-DOAS data ($R^2 = 0.75$) and shows a small normalized mean bias (NMB = 0.8 %). The reduced major axis (RMA) regression shows a slope of 0.70, mainly because of the underestimate on high-NO$_2$ days. TM5-MP-DOMINO is also correlated with MAX-DOAS ($R^2 = 0.68$), although the correlation is weaker than our retrieval. The NMB of TM5-MP-DOMINO is much more significant ($-41.9$ %), and the RMA regression slope is much smaller (0.42). Similar underestimates of TM5-MP-DOMINO have been discussed in their "Readme" document (Eskes et al., 2019) and the algorithm theoretical basis document (ATBD) file (van Geffen et al., 2019) in general, in Griffin et al. (2019) for Canada. Major plausible causes of the underestimate of TM5-MP-

**Table 1.** MAX-DOAS measurement sites.

| Site name | Geographical location | Measurement time |
|---|---|---|
| Nanjing | 118.950° E, 32.118° N, 36 m | 1 July–31 October 2018 |
| Xuzhou | 117.142° E, 34.217° N, 92 m | 1 July–31 October 2018 |
| Fukue | 128.680° E, 32.750° N, 83 m | 1 July–15 September 2018 |

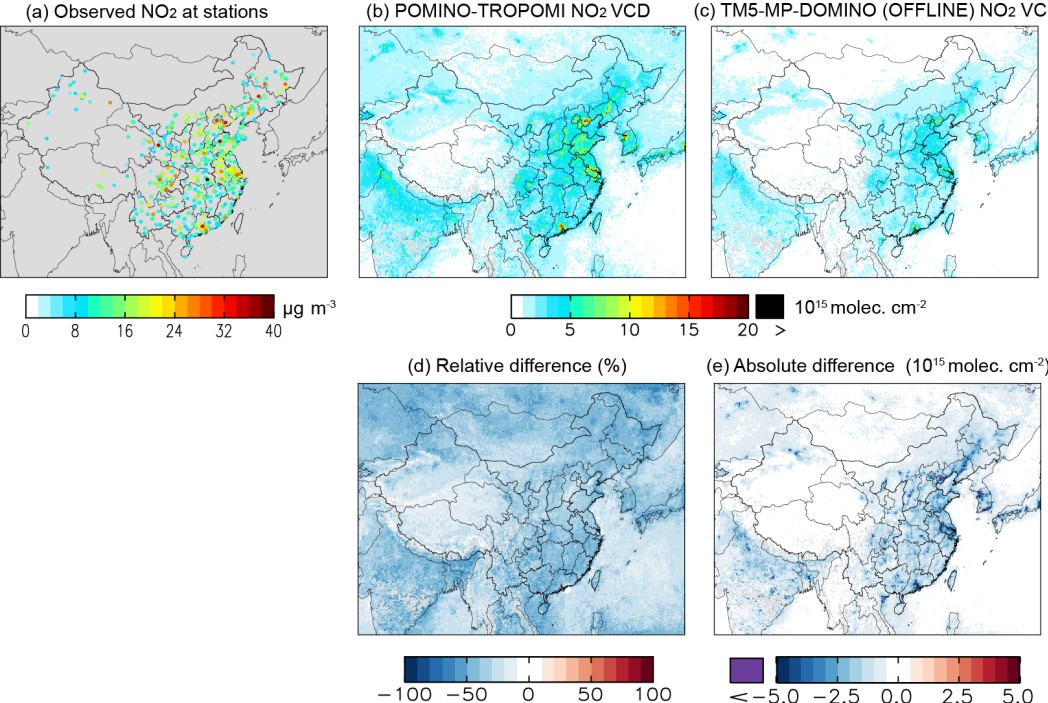

**Figure 4.** The spatial distribution of (**a**) near-surface NO$_2$ concentrations at monitoring stations, (**b**) POMINO-TROPOMI NO$_2$ VCD, and (**c**) TM5-MP-DOMINO (OFFLINE) NO$_2$ VCD at 0.05° × 0.05° grid in July 2018. Panels (**d**) and (**e**) are relative and absolute difference of TM5-MP-DOMINO (OFFLINE) to POMINO-TROPOMI NO$_2$ VCD.

DOMINO include coarse-resolution climatological surface albedo data, coarse-resolution (1° × 1°) a priori NO$_2$ profiles, implicit aerosol corrections, and uncertainties in CP from FRESCO-S.

Cloud pressures from FRESCO-S are found to be too high – i.e., the cloud top is too close to the ground, especially over China (van Geffen et al., 2019). We examine this effect by excluding the pixels with CP > 850 hPa when comparing with MAX-DOAS data. With this additional criterion, the number of valid days drops to 20. Figure 6c and d show the scatterplots and corresponding RMA results. The NMB of TM5-MP-DOMINO is reduced slightly to −38.2 %, and its $R^2$ for day-to-day variation is increased from 0.68 to 0.85. POMINO-TROPOMI still outperforms TM5-MP-DOMINO: 0.85 versus 0.85 for $R^2$, 13.8 % versus −38.2 % for NMB, and 0.82 versus 0.56 for RMA regression slope. The improvement from excluding CP > 850 hPa scenes is larger in TM5-MP-DOMINO (with implicit aerosol corrections) than in POMINO-TROPOMI

(with explicit aerosol corrections). The averaged CF of data excluding CP > 850 hPa is 0.13 (AOD = 0.63), which is much larger than the averaged value (CF = 0.06) in Fig. 6a and b (AOD = 0.57). This appears to imply that the overestimated CP may be partly because the FRESCO-S cloud algorithm might misinterpret heavy aerosol loadings near the ground as clouds, a common issue in satellite data (Lin and Li, 2016).

## 3.2 Influences of aerosol correction approaches and horizontal resolutions of a priori NO$_2$ profiles

To investigate the causes of difference between POMINO-TROPOMI and TM5-MP-DOMINO (OFFLINE), we conduct two sensitivity retrievals based on the POMINO-TROPOMI algorithm (Cases 1 and 2 in Table 2). Figure 7 shows the relative (panels a–c) and absolute (panels d–f) differences between retrieval cases (REF, Case 1, and Case 2). Case 1 adopts the implicit aerosol correction for both clouds and NO$_2$ retrievals, as in TM5-MP-DOMINO, so the dif-

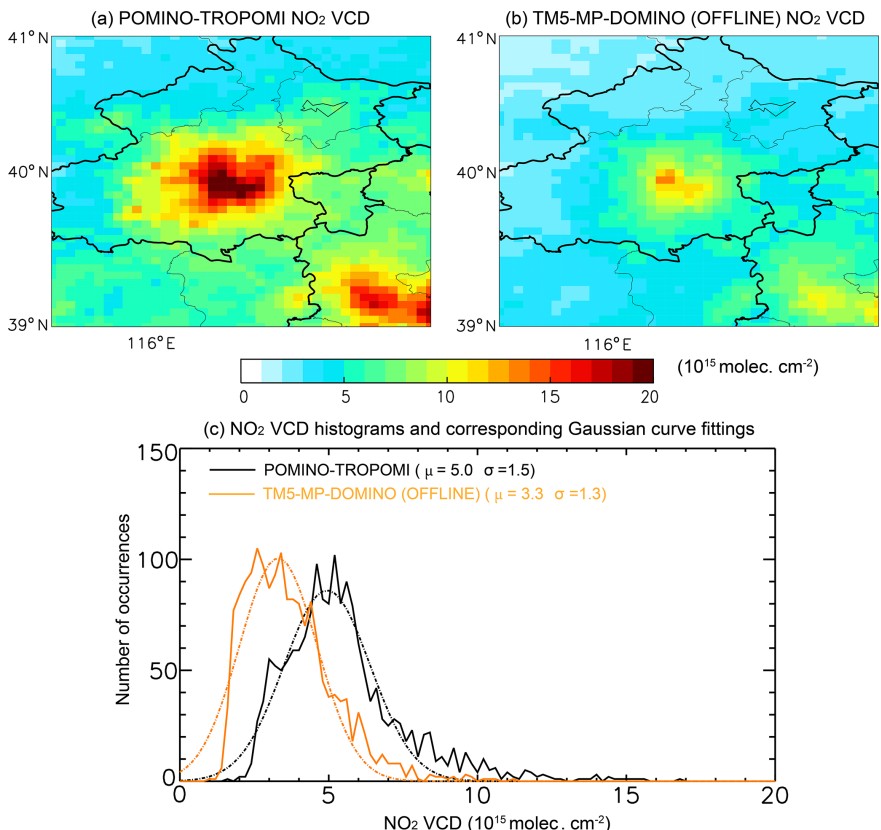

**Figure 5.** Spatial distributions of **(a)** POMINO-TROPOMI NO$_2$ VCDs and **(b)** TM5-MP-DOMINO (OFFLINE) NO$_2$ VCDs on a 0.05° × 0.05° grid over Beijing and its surrounding areas averaged over July 2018. **(c)** Histograms of monthly NO$_2$ VCDs over this region. The bin size is $0.2 \times 10^{15}$ molec. cm$^{-2}$. The black and orange dashed lines are corresponding Gaussian curve fitting of the histograms. μ is the mean value, and σ is the standard deviation of a Gaussian curve fitting.

ference between REF and Case 1 indicates the effect of aerosol representation (explicit versus implicit) (Fig. 7a, d). In Case 2, the high-resolution (0.25° lat × 0.3125° long) NO$_2$ profiles are replaced by low-resolution (2° lat × 2.5° long) profiles simulated by the GEOS-Chem global model; aerosols are represented implicitly as in Case 1. Case 2 thus mimics TM5-MP-DOMINO, which uses an implicit aerosol correction and coarse-resolution NO$_2$ profiles. Thus, the difference between Case 1 and Case 2 arises from the a priori NO$_2$ profiles (Fig. 7b, e). The difference between Case 2 and REF further indicates the joint effect of using coarse-resolution a priori NO$_2$ profiles and an implicit aerosol correction (Fig. 7c, f).

Figure 7 shows that the individual influences of aerosol representations (explicit versus implicit) and a priori NO$_2$ profiles (fine versus coarse resolution) vary substantially from one location to another. The impacts of aerosol corrections are most evident over the areas of heavy aerosol loadings including East China, India, and parts of Southeast Asia. The implicit aerosol correction (Case 1) tends to result in lower NO$_2$ VCDs by 0 %–50 % over urban areas compared to the explicit aerosol correction (Case REF) (Fig. 7a, d). By

comparison, the impacts of NO$_2$ profiles are spatially more heterogeneous (Fig. 7b and e). Over the offshore coastal areas, using coarse-resolution NO$_2$ profiles (Case 2) tends to overestimate the NO$_2$ VCDs by 30 %–100 % relative to when high-resolution profiles are used (Case 1), due to (horizontal) over-smoothing of NO$_2$.

Below, we discuss these differences over two key areas including northern East China and Xinjiang. Northern Eastern China (bounded by the black rectangle in Fig. 7a) is a region with heavy aerosol loading region (Fig. 2). Over this region, an implicit rather than explicit aerosol representation results in lower NO$_2$ VCDs by ∼ 25 % (Fig. 7a, d), while the effect of NO$_2$ profiles is weaker (Fig. 7b, e). The joint effect is dominated by the effect of aerosol representation (Fig. 7c, f).

That the impact of a priori NO$_2$ profiles is relatively small over northern Eastern China is partly because of the smearing effect by wind. Figure 8 differentiates the effects of NO$_2$ profiles between 23 windy (daily average wind speed under 500 m > 2 m s$^{-1}$) days and 7 calm (wind speed < 2 m s$^{-1}$) days. The dataset of wind is taken from the National Aeronautics and Space Administration–Global Modeling and Assimilation Office's (NASA–GMAO's) "forward-processing"

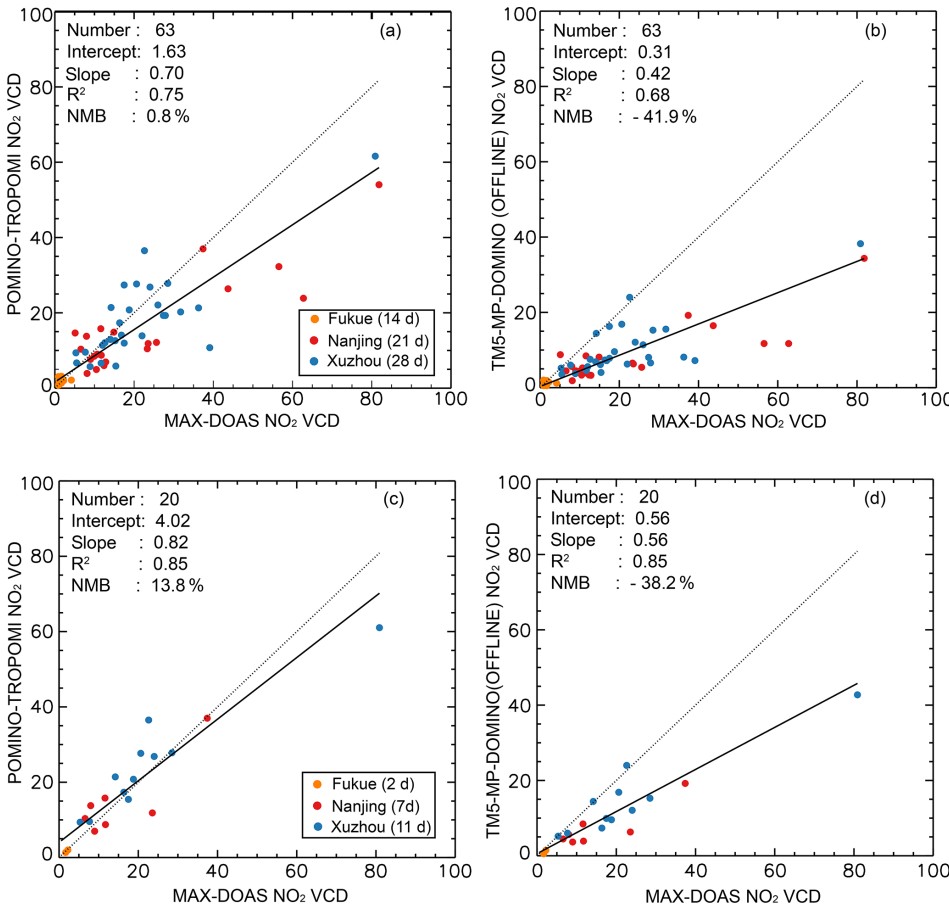

**Figure 6.** Scatterplot for daily NO$_2$ VCDs ($10^{15}$ molec. cm$^{-2}$) between MAX-DOAS and two TROPOMI NO$_2$ VCD products. Each colored dot represents a day, and each color denotes a station. For each day, the satellite data are averaged over all pixels. Panels **(c)** and **(d)** are results of two TROPOMI NO$_2$ VCD products with effective cloud pressures $\leq 850$ hPa.

**Table 2.** Sensitivity experiments for the NO$_2$ retrieval based on the POMINO-TROPOMI algorithm (n/a: not applicable). [TS3]

| ID | A priori NO$_2$ profiles | Aerosols | Surface reflectance |
|---|---|---|---|
| Case REF (POMINO-TROPOMI) | 0.25° lat × 0.3125° long | Explicit | MODIS BRDF |
| Case 1 | Same as Case REF | n/a | Same as Case REF |
| Case 2 | 2° lat × 2.5° long | n/a | Same as Case REF |
| Case 3 | Same as Case REF | Semi-explicit[1] | Same as Case REF |
| Case 4 | Same as Case REF | Same as Case REF | OMI LER[2] |

[1] Explicit aerosol treatments for $M_{clr}$ and no aerosol corrections for $M_{cld}$. [2] The LER dataset is a 5-year climatology built upon OMI measurements on a grid of 0.5° × 0.5°. The dataset is taken from the TM5-MP-DOMINO product.

(GEOS-FP) data product with the horizontal resolution at 0.25° lat × 0.3125° long. In the windy cases, the NO$_2$ VCDs are much more smoothed even at 0.3125° resolution, and thus the difference of NO$_2$ resolutions is very small. In contrast, NO$_2$ VCDs exhibit much stronger horizontal gradient in calm situations, which are better retrieved when high-resolution NO$_2$ profiles are used. As calm situation is more favorable for pollutant accumulation, while windy days help

to dilute concentration of NO$_2$, so much higher NO$_2$ VCDs are found in Fig. 8a and b.

Xinjiang (bounded by the red rectangle in Fig. 7a) is a deserted region in West China. Over Xinjiang, the resolution of a priori NO$_2$ profiles affects the retrieved NO$_2$ VCDs much more than the aerosol representation does (Fig. 7b and e). Compared to Case 1 (with high-resolution NO$_2$ profiles), Case 2 (with coarse-resolution profiles) leads to much lower

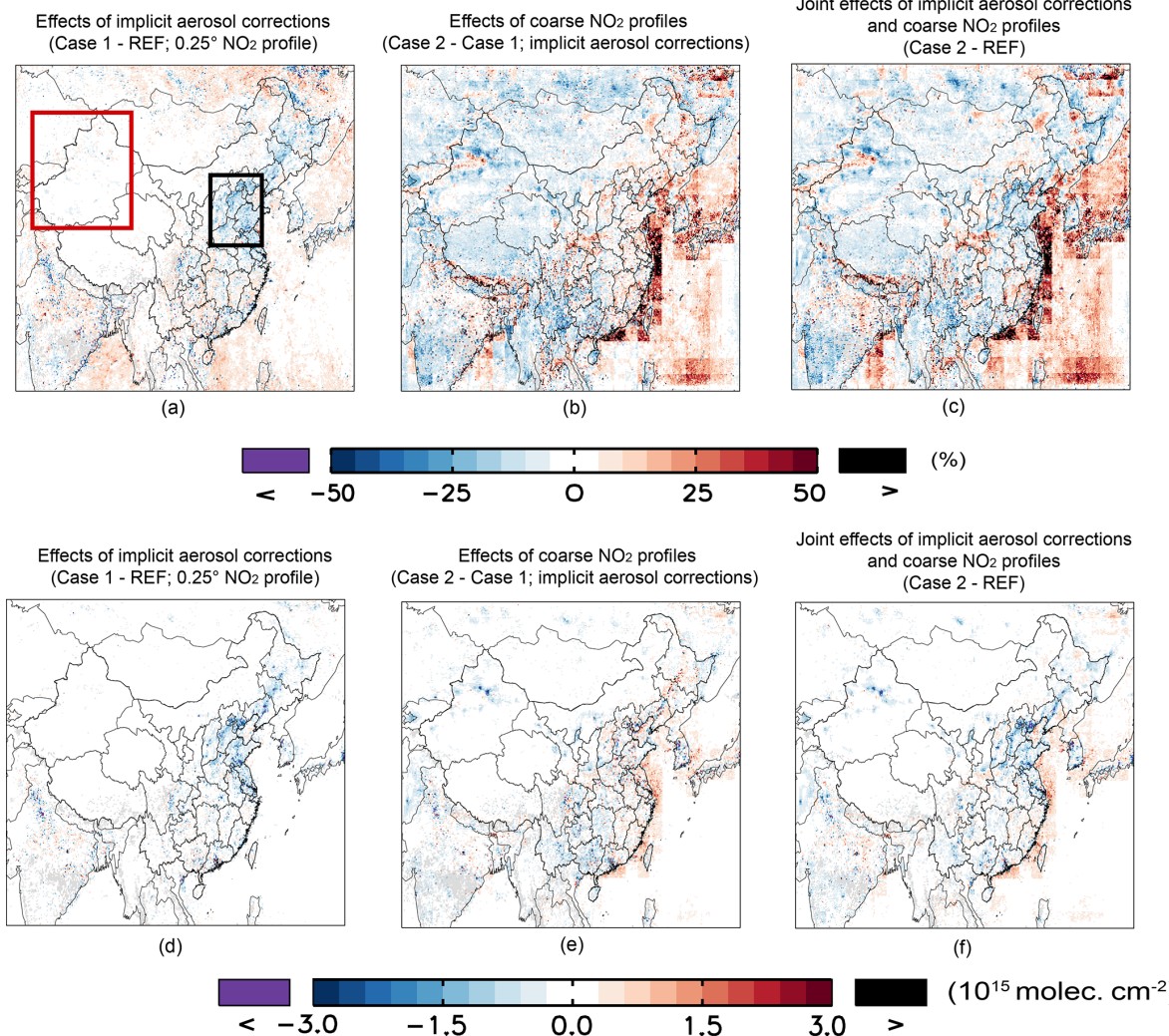

**Figure 7. (a–c)** Relative differences caused by aerosol corrections and a priori NO$_2$ profiles. Panels **(d)–(f)** are the corresponding absolute differences. The black and red rectangles stand for northern Eastern China and Xinjiang, respectively.

NO$_2$ VCDs over the isolated urban areas which are not resolved by the coarse model.

## 3.3 Influences of directly using the CP data from FRESCO-S

As we take the CP data directly from the FRESCO-S retrieval rather than re-retrieving CP (as done for CF), two main issues arise. First, the FRESCO-S retrieved CP may be affected by aerosols, and thus using such CP data in our explicit aerosol corrections (Case REF) may lead to overcorrection of aerosol effects. To estimate the effect of such overcorrection on retrieved NO$_2$ VCDs, we employ in an additional sensitivity case (Case 3 in Table 2) a "semi-explicit" aerosol correction approach. This approach explicitly includes aerosols in the calculation of AMF for the clear-sky portion ($M_{clr}$) of a pixel (as in Case REF) but excludes aerosols for the cloudy-sky portion ($M_{cld}$) of that pixel. Correspondingly, CF is recalcu-

lated on the basis that the radiance at 437.5 nm received by TROPOMI is contributed from the aerosol-contained clear-sky portion and the no-aerosol, cloudy-sky portion. Table 3 shows that in July 2018, on a pixel basis, the derived NO$_2$ profiles in Case 3 are larger than those in Case REF, with an average difference increasing from 3.1 % at relatively clean situations (NO$_2$ VCDs in Case REF $< 5 \times 10^{15}$ molec. cm$^{-2}$) to 11.2 % for polluted situations (NO$_2$ VCDs in Case REF $\geq 15 \times 10^{15}$ molec. cm$^{-2}$). The spatial distributions in Fig. S1a and S1b also show higher NO$_2$ VCDs in Case 3 relative to Case REF. The corresponding increases in CF (Fig. S1c versus Fig. S1d) are because in Case 3 the scattering contributions to the radiance from aerosols in the cloudy-sky portion (that would have occurred) are accounted for with higher CFs. The enhanced "shielding effect" of clouds (due to higher CFs) result in lower NO$_2$ AMFs and higher VCDs.

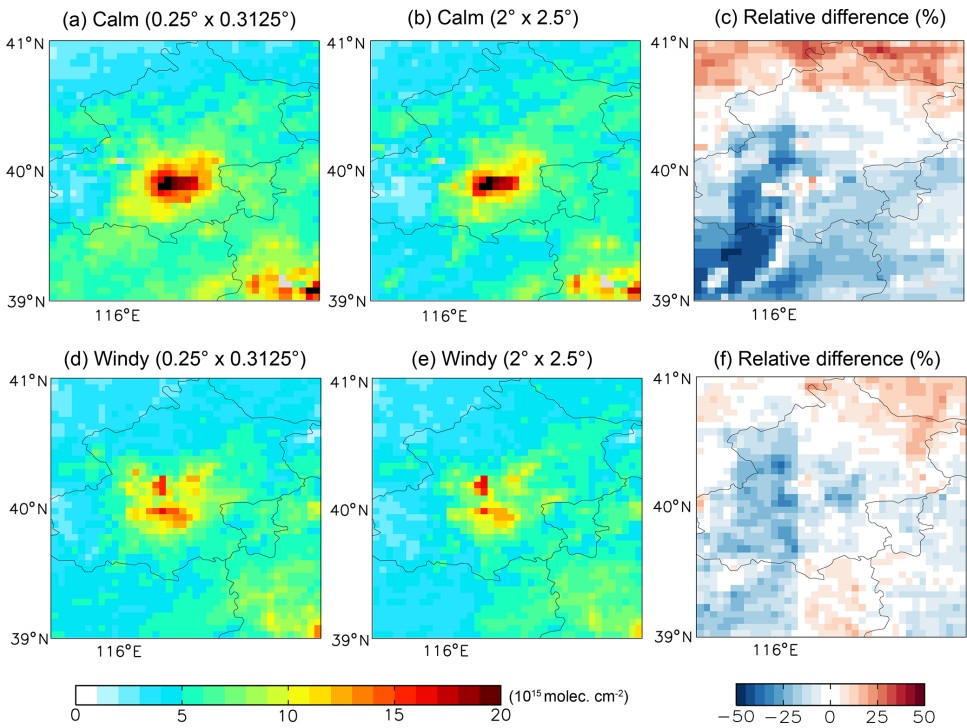

**Figure 8.** Spatial distributions of POMINO-TROPOMI NO$_2$ VCDs on a $0.05° \times 0.05°$ grid under calm conditions with **(a)** $0.25°$ lat $\times 0.3125°$ long NO$_2$ profiles (Case 1), with **(b)** $2°$ lat $\times 2.5°$ long profiles (Case 2), and **(c)** their relative differences. Panels **(d)**–**(f)** are similar to panels **(a)**–**(c)** but under windy conditions.

**Table 3.** Effects of choices of aerosols and surface reflectance inconsistent with using the CP data from FRESCO-S in July 2018

| Situation[a] | Effect of aerosol correction choice (Case 3 – REF)[b] | Effect of surface reflectance choice (Case 4 – REF)[b] |
|---|---|---|
| NO$_2$ VCD $< 5 \times 10^{15}$ molec. cm$^{-2}$<br>Mean AOD: 0.27<br>Mean CF: 0.07<br>Mean CP: 748 hPa | 3.1 % | −3.7 % |
| $5 \times 10^{15}$ molec. cm$^{-2} \leq$ NO$_2$ VCD<br>$< 15 \times 10^{15}$ molec. cm$^{-2}$<br>Mean AOD: 0.65<br>Mean CF: 0.08<br>Mean CP: 767 hPa | 4.1 % | −8.1 % |
| $15 \times 10^{15}$ molec. cm$^{-2} \leq$ NO$_2$ VCD<br>Mean AOD: 0.66<br>Mean CF: 0.09<br>Mean CP: 772 hPa | 11.2 % | −8.3 % |

[a] The values of AOD, CF, and CP shown here are mean values of the pixels of corresponding subsets in Case REF. [b] The percentage values represent the mean relative differences relative to Case REF.

For surface reflectance, Case REF considers the BRDF effect instead of Lambertian equivalent reflectivity (LER) which is used in deriving FRESCO-S clouds. This leads to inconsistency between when the CP is derived and when it is used. The LER data used by FRESCO-S are gener-

ated at 758 and 772 nm (based on the Global Ozone Monitoring Experiment-2), rather than at the 437.5 nm for the NO$_2$ retrieval. Thus Case 4 adopts the OMI LER data from TM5-MP-DOMINO – a 5-year, monthly-based climatology at 440 nm – and recalculates CFs and NO$_2$ AMFs with ex-

**Table 4.** Contributions of estimated errors in POMINO-TROPOMI $NO_2$ AMFs.

| Error source | | Estimated error in parameter | Corresponding error in tropospheric $NO_2$ AMF |
|---|---|---|---|
| $NO_2$ profiles | | | $\pm 10\%$[a] |
| Cloud fraction | | $\pm 0.01$[b] | $\pm 10\%$[b] |
| Cloud pressure | | $\pm 50\,hPa$[c] | $\pm 10\%$[c] |
| BRDF coefficients | | $\pm 10\%$[d] | $\pm 10\%$[d] |
| Aerosol | AOD | $\pm 0.07$[e] | $\pm 10\%$ for clean situations |
| | SSA | $\pm 0.03$[f] | Up to $\pm 20\%$ for heavily polluted |
| | Aerosol layer height | Underestimated by 300–600 m[g] | cases[h] |

[a] This error estimate is based on Lin et al. (2010), Lin et al. (2014), Boersma et al. (2004, 2011, 2018) and this study. The error accounts for the effect of horizontal resolutions and the vertical process in GEOS-Chem. [b] This accounts for the expectation that our explicit aerosol correction leads to more reasonable CFs. [c] Based on van Geffen et al. (2019). Our estimated $NO_2$ errors related to the use of FRESCO CP data (instead of recalculating it) are within this error range. [d] The estimated parameter error is taken from Zhou et al. (2010), and the corresponding AMF error is based on Lin et al. (2014) and Case 4. [e] Based on Tian et al. (2018), which compared the MODIS Merged AOD C6.1 product with AERONET in China. [f] Based on Lin et al. (2015), which compared GEOS-Chem simulated SSA with Lee et al. (2007). [g] Based on Liu et al. (2019), which compared GEOS-Chem simulated aerosol extinction profiles with CALIOP. [h] This is a tentative error estimate based on Lin et al. (2015), Lorente et al. (2017), Liu et al. (2019), and this study. With explicit aerosol corrections, the errors in heavily polluted situations are expected to be smaller than when assuming implicit aerosol corrections.

plicit aerosol corrections and FRESCO-S CP (Table 2). Here, the ice/snow flag in the TM5-MP-DOMINO product is used to exclude the possible ice/snow contamination, and only the pixels with blue-sky albedos (derived from the BRDF data in Case REF) less than 0.3 are taken into consideration. The resulting $NO_2$ VCDs in Case 4 are lower than Case REF by 3.7 % on a pixel-based average for relatively clean situations and by 8.3 % for polluted situations (Table 3). Figure S2a and S2b further show the spatial distributions of the relative and absolute differences in derived monthly mean $NO_2$ VCDs between Case 4 and Case REF. In general, Case 4 leads to lower $NO_2$ VCDs than Case REF because of stronger surface reflectance, as is obvious in the comparison of blue-sky albedo in Case REF and LER albedo in Case 4 (Fig. S2c versus Fig. S2d).

### 3.4 Error estimates for POMINO-TROPOMI $NO_2$ AMFs

It is difficult to derive an overall AMF error for each pixel with our algorithm, particularly because of the three-dimensional aerosol parameters used, the interlinkage between clouds and other parameters (aerosols and surface reflectance), and the exclusion of lookup tables (LUTs; which leads to too computationally expensive error estimates). Table 4 provides a preliminary estimate of the $NO_2$ AMF errors with respect to uncertainties in individual parameters. We follow the ATBD of TM5-MP-DOMINO (van Geffen et al., 2019) and make use of error estimates in previous studies. Individual errors with respect to CF, CP, and BRDF coefficients are within 10 %. Errors with respect to aerosols are consid-

ered to be larger, due to uncertainty in AOD, single-scattering albedo (SSA), and aerosol vertical profiles (Liu et al., 2019), as well as the fact that using the CP from FRESCO-S rather than deriving it here may lead to an additional error in the $NO_2$ AMFs. Note that these individual errors are not fully independent due to the aforementioned linkage between parameters. The magnitude of potential systematic bias in $NO_2$ may be smaller than the quadrature sum of errors in individual parameters, as suggested by the slight mean bias relative to MAX-DOAS data (Fig. 6a).

### 4 Conclusion and discussion

The POMINO algorithm to retrieve tropospheric $NO_2$ VCDs has been successfully applied to TROPOMI over East Asia. The resulting POMINO-TROPOMI product shows higher tropospheric $NO_2$ VCDs (by about 35 % averaged over East Asia) and much clearer urban and other hotspot signals, compared to the TM5-MP-DOMINO (OFFLINE) product. Further evaluation using independent MAX-DOAS measurements indicates very good performance of POMINO-TROPOMI in capturing the day-to-day variation ($R^2 = 0.75$; $N = 63$) and mean value (NMB = 0.8 %) of $NO_2$, better than TM5-MP-DOMINO (0.68 and $-41.9$ %, respectively).

Over regions with heavy aerosol loading, the accuracy of retrieved $NO_2$ VCDs is affected substantially by how aerosols are represented in the retrieval process (implicit or explicit). The implicit aerosol representation tends to underestimate the $NO_2$ VCDs by 0 %–50 % over most urban areas in East Asia and by about 25 % over northern East

China. Using a priori NO$_2$ profile data at a horizontal resolution of $\sim 25$ km, POMINO-TROPOMI captures the city-scale NO$_2$ hotspots. Reducing the horizontal resolution of a priori profiles to what is typically set up by global models (100–200 km) underestimates the NO$_2$ hotspots and the spatial gradient surrounding them, and the effects are more pronounced in calm than in windy situations. Overall, our results provide useful information to improve TROPOMI retrieval algorithms and offer insight for applications to the upcoming geostationary satellite instruments including GEMS, TEMPO, and Sentinel-4 Precursor.

Further work can be done to improve the retrieval algorithm for TROPOMI. First, the hybrid cloud retrieval method used in POMINO-TROPOMI is not optimal, because only cloud fraction but not cloud pressure is recalculated with explicit aerosol corrections and BRDF effects. The uncertainty caused by inconsistent assumptions of aerosols and albedos in cloud pressure and NO$_2$ VCD retrievals are initially estimated in this study. Recalculation of cloud pressure will be done in the future using the O$_2$–O$_2$ method when the O$_2$–O$_2$ SCD data are available. Second, correcting the simulated aerosol extinction vertical profiles will further improve the clouds and NO$_2$ retrievals (Liu et al., 2019). Third, the horizontal resolution of a priori NO$_2$ profiles ($\sim 25$ km) does not match the fine footprint of TROPOMI, and further increasing the resolution will help achieve better accuracy for analyses of very fine scale pollution characteristics such as along the highways and rivers and within urban centers.

*Data availability.* The POMINO-TROPOMI NO$_2$ data are available at the ACM group product website (http://www.pku-atmos-acm.org/acmProduct.php/, Lin et al., 2020). The TM5-MP-DOMINO product can be download via the TEMIS website at http://www.temis.nl/airpollution/no2.html Royal Netherlands Meteorological Institute, 2020). The near-surface data of NO$_2$ and PM$_{2.5}$ can be downloaded from http://www.cnemc.cn/sssj/cskqzl/ (Ministry of Ecology and Environment, 2019). The ground-based MAX-DOAS observations would be provided after the applications of users are approved by corresponding owners.

*Supplement.* The supplement related to this article is available online at: https://doi.org/10.5194/amt-13-1-2020-supplement.

*Author contributions.* JL conceived the research. ML and JL designed the experiment. ML performed all calculations with some code support from HK and HW. ML and JL wrote the paper with inputs from KFB and HE. RS provided LIDORT. YK, QH, XT, KQ, and PX provided the MAX-DOAS observations. RN helped to process surface observations. YY and JW helped to analyze the relationship between meteorological and NO$_2$ VCD variations. All authors commented on the paper.

*Competing interests.* The authors declare that they have no conflict of interest.

*Special issue statement.* This article is part of the special issue "TROPOMI on Sentinel-5 Precursor: first year in operation (AMT/ACP inter-journal SI)". It is not associated with a conference.

*Financial support.* This research has been supported by the National Natural Science Foundation of China (grant no. 41775115) and the second Tibetan Plateau Scientific Expedition and Research Program (grant no. 2019QZKK0604).

*Review statement.* This paper was edited by Ben Veihelmann and reviewed by two anonymous referees.

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

**Remarks from the typesetter**

TS1    The additional comma has been removed.

TS2    Please note that it is our standard to cite several citations from the same author like this.

TS3    Please note that exchanging Tables 1 and 2 requires the editor's approval. Please provide a detailed explanation for this change that can be forwarded to the editor. Please note that this entire process will be available online after publication. Upon approval, we will make the appropriate changes. Thank you for your understanding.