# Peer review of "A new TROPOMI product for tropospheric NO2 columns over East 2 Asia with explicit aerosol corrections"

_Atmospheric Measurement Techniques, 2019_

## Referee Comment (RC1) · Anonymous Referee #1 · 1 Mar 2020

General comments:

This manuscript describes a tropospheric NO2 column retrieval algorithm applied to the new sensor TROPOMI. Results are validated using ground-based measurements, and the impact of introducing an explicit aerosol correction and improving the horizontal resolution of a priori NO2 profile shapes used in the retrieval are discussed.

This study is based on the author's previous work on OMI tropospheric NO2 retrievals using the POMINO algorithm. A simplified version of this algorithm is developed here and applied to the new TROPOMI sensor. The main simplification concerns the cloud pressure retrieval, which no longer accounts for aerosols, while an explicit aerosol correction is maintained for the NO2 retrieval itself. The impact of this simplification should be more extensively discussed and accounted for in the error analysis. In addition, the

aerosol correction is treated twice in the product (explicit correction for the NO2 retrieval, and implicit correction for the cloud pressure retrieval), which may lead to larger retrieval errors than more simple treatments (e.g. only explicit aerosol correction without cloud, or simple cloud correction including implicit aerosol correction). Therefore, I strongly encourage the authors to expand their discussion and provide more robust and convincing arguments in support of their approach. I recommend publication of this manuscript only after major revision addressing the aforementioned comments.

Specific comments:

Figure 1: Reflectance at 758nm -> 437.5nm.

Figures 6-7: these figures can be merged. I suggest to add similar figures where the y-axis use retrieved MAX-DOAS profiles smoothed by satellite averaging kernels. This will remove the error on the satellite retrieval coming from the profile shape uncertainty.

Page 17 line 9: more much -> much more

SI Table S2: Number of measurements within +/- 0.5h -> Number of measurements within +/- 1h, why?

SI reference is missing.

---

## Referee Comment (RC2) · Anonymous Referee #2 · 15 Apr 2020

Major comments:

Liu et al. reported an improved tropospheric NO2 retrieval over East Asia by accounting for the effect of aerosol and using a priori NO2 profile from high-resolution chemical model. These results are generally showing better agreements compared to the ground-based MAX-DOAS measurements. As an alternative NO2 data product over East Asia (compared to the operational product), the manuscript will benefit significantly from additional assessment on the NO2 uncertainty which are of great importance to the user community. Also, The following major concerns should be well addressed before the final publication in the AMT journal.

1. The reviewer #1 has concerned on the inconsistent treatment of aerosol corrections between the NO2 AMFs calculation and cloud pressure retrieval. In fact, there are

many conflicting considerations due to the hybrid POMINO algorithm itself (i.e., using tropospheric NO2 SCDs and cloud pressure from the operational product). For example, the surface BRDF parameters are used in the tropospheric NO2 AMF calculations, while Lambertian Equivalent Reflectance (LER) are assumed during the retrieval of cloud pressure and tropospheric NO2 SCDs. These conflicting considerations may introduce larger error sources on the NO2 retrieval compared to other consistent independent retrieval algorithm. It should be investigated and discussed in the manuscript.

2. The manuscript lacks of necessary uncertainty estimation on tropospheric NO2 retrieval, considering the existing conflicting considerations in the hybrid retrieval algorithm. Also, a reasonable error budget is beneficial to the potential users.

3. In Fig. 6, only limited MAX-DOAS observations on Summer 2018 (N=63 ) are used to validate satellite NO2 retrieval. However, aerosol consideration can make larger difference to NO2 retrieval especially in winter with severe haze pollution. It would be more convincing to include more data in temporal coverage.

Specific comments:

1. Page 4, Line 8-9: compared to the previous POMINO version in Liu et al., 2019, constraining aerosol profile with CALIPSO data is no longer used in this paper. Why? It would be better to explain these algorithm changes.

---

## Author Comment (AC1) · 30 May 2020

General comments:

1, The main simplification concerns the cloud pressure retrieval, which no longer accounts for aerosols, while an explicit aerosol correction is maintained for the NO2 retrieval itself. The impact of this simplification should be more extensively discussed and accounted for in the error analysis. In addition, the aerosol correction is treated twice in the product (explicit correction for the NO2 retrieval, and implicit correction for the cloud pressure retrieval), which may lead to larger retrieval errors than more simple treatments (e.g. only explicit aerosol correction without cloud, or simple cloud correction including implicit aerosol correction). Therefore, I strongly encourage the authors
to expand their discussion and provide more robust and convincing arguments in support of their approach. I recommend publication of this manuscript only after major revision addressing the aforementioned comments.

Thank you very much for your suggestions. We have added a section (Section 3.3) and additional experiments to test the effect of such issue.

In Line 161-165, we add:"

Two additional cases (Cases 3 and 4) concern the treatment of CP in combination with the choice of aerosols and surface reflectance. Specifically, using the CP data directly from FRESCO-S means that our retrieval algorithm does not perfectly account for the effect of aerosols on clouds. Our retrieval consider the BRDF effects while Lambertian surface is used in deriving the FRESCO-S CP. "

In Line 347-385, we add:"

3.3 Influences of directly using the CP data from FRESCO-S

As we take the CP data directly from the FRESCO-S retrieval rather than re-retrieving CP (as done for CF), two main issues arise. First, the FRESCO-S retrieved CP may be affected by aerosols, thus using such CP data in our explicit aerosol corrections (Case REF) may lead to over-correction of aerosol effects. To estimate the effect of such over-correction on retrieved NO2 VCDs, we employ in an additional sensitivity case (Case 3 in Table 2) a "semi-explicit" aerosol correction approach. This approach explicitly includes aerosols in the calculation of AMF for the clear-sky portion (Mclr) of a pixel (as in Case REF), but excludes aerosols for the cloudy-sky portion (Mcld) of that pixel. Correspondingly, CF is re-calculated on the basis that the radiance at 437.5 nm received by TROPOMI is contributed from the aerosol-contained clear-sky portion and the no-aerosol cloudy-sky portion. Table 3 shows that in July 2018, on a pixel basis, the derived NO2 in Case 3 are larger than those in Case REF, with an average difference increasing from 3.1% at relatively clean situations (NO2 VCDs in Case REF

< 5 × 10ˆ15 molecu·cmˆ(-2)) to 11.2% for polluted situations (NO2 VCDs in Case REF ≥ 15 × 10ˆ15 molecu·cmˆ(-2)). The spatial distributions in Fig. S1a and S1b also show higher NO2 VCDs in Case 3 relative to Case REF. The corresponding increases in CF (Fig. S1c versus S1d) are because in Case 3 the scattering contributions to the radiance from aerosols in the cloudy-sky portion (that would have occurred) are accounted for with higher CFs. The enhanced "shielding effect" of clouds (due to higher CFs) result in lower NO2 AMFs and higher VCDs.

For surface reflectance, Case REF considers the BRDF effect instead of Lambertian Equivalent Reflectivity (LER) which is used in deriving FRESCO-S clouds. This leads to inconsistency between when the CP is derived and when it is used. The LER data used by FRESCO-S are generated at 758 and 772 nm (based on the Global Ozone Monitoring Experiment-2), rather than at the 437.5 nm for the NO2 retrieval. Thus Case 4 adopts the OMI LER data from TM5-MP-DOMINO, a five-year monthly based climatology at 440 nm, and re-calculates CFs and NO2 AMFs with explicit aerosol corrections and FRESCO-S CP (Table 2). Here, the ice-snow flag in the TM5-MP-DOMINO product is used to exclude the possible ice/snow contamination, and only the pixels with blue-sky albedos (derived from the BRDF data in Case REF) less than 0.3 are taken into consideration. The resulting NO2 VCDs in Case 4 are lower than Case REF by 3.7% on pixel-based average for relatively clean situations and by 8.3% for polluted situations (Table 3). Figure S2a and S2b further shows the spatial distributions of the relative and absolute differences in derived monthly mean NO2 VCDs between Case 4 and Case REF. In general, Case 4 leads to lower NO2 VCDs than Case REF because of stronger surface reflectance, as is obvious in the comparison of blue-sky albedo in Case REF and LER albedo in Case 4 (Fig. S2c versus S2d). "

Specific comments:

1, Figure 1: Reflectance at 758nm -> 437.5nm.

Changed.

[Figure]

2, Figures 6-7: these figures can be merged. I suggest to add similar figures where the y-axis use retrieved MAX-DOAS profiles smoothed by satellite averaging kernels. This will remove the error on the satellite retrieval coming from the profile shape uncertainty.

Figure 7 has been merged with Fig. 6.

It is a good suggestion to show the results using retrieved MAX-DOAS profiles smoothed by satellite averaging kernels. However, we do not have MAX-DOAS NO2 profiles at any station.

3, Page 17 line 9: more much -> much more

Changed.

4, SI Table S2: Number of measurements within +/- 0.5h -> Number of measurements within +/- 1h, why?

This phenomenon occurs at Xuzhou. It is because our criterion to ensure weak temporal variability of NO2 within the time window; large variability would suggest very local signals that can contaminate the comparison. Indeed, expanding the sampling (temporal) window leads to much stronger variability at Xuzhou, leading to fewer numbers of valid days (7 days are excluded in Table S2).

In Line 185-187, we stated that "To reduce the influence of local events, we exclude all MAX-DOAS data whose standard deviations within the period exceed 20% of their mean values.". We have added a footnote in Table S2 to explain this issue.

5, SI reference is missing.

Added.

Please also note the supplement to this comment:
https://www.atmos-meas-tech-discuss.net/amt-2019-500/amt-2019-500-AC1-supplement.pdf

**Supplement:**

**General comments:**

The main simplification concerns the cloud pressure retrieval, which no longer accounts for aerosols, while an explicit aerosol correction is maintained for the $NO_2$ retrieval itself. The impact of this simplification should be more extensively discussed and accounted for in the error analysis. In addition, the aerosol correction is treated twice in the product (explicit correction for the NO2 retrieval, and implicit correction for the cloud pressure retrieval), which may lead to larger retrieval errors than more simple treatments (e.g. only explicit aerosol correction without cloud, or simple cloud correction including implicit aerosol correction). Therefore, I strongly encourage the authors to expand their discussion and provide more robust and convincing arguments in support of their approach. I recommend publication of this manuscript only after major revision addressing the aforementioned comments.

Thank you very much for your suggestions. We have added a section (Section 3.3) and additional experiments to test the effect of such issue.

In Line 161-165, we add:"

Two additional cases (Cases 3 and 4) concern the treatment of CP in combination with the choice of aerosols and surface reflectance. Specifically, using the CP data directly from FRESCO-S means that our retrieval algorithm does not perfectly account for the effect of aerosols on clouds. Our retrieval consider the BRDF effects while Lambertian surface is used in deriving the FRESCO-S CP. "

In Line 347-385, we add:"

*3.3 Influences of directly using the CP data from FRESCO-S*

As we take the CP data directly from the FRESCO-S retrieval rather than re-retrieving CP (as done for CF), two main issues arise. First, the FRESCO-S retrieved CP may be affected by aerosols, thus using such CP data in our explicit aerosol corrections (Case REF) may lead to over-correction of aerosol effects. To estimate the effect of such over-correction on retrieved $NO_2$ VCDs, we employ in an additional sensitivity case (Case 3 in Table 2) a "semi-explicit" aerosol correction approach. This approach explicitly includes aerosols in the calculation of AMF for the clear-sky portion ($M_{clr}$) of a pixel (as in Case REF), but excludes aerosols for the cloudy-sky portion ($M_{cld}$) of that pixel. Correspondingly, CF is re-calculated on the basis that the radiance at 437.5 nm received by TROPOMI is contributed from the aerosol-contained clear-sky portion and the no-aerosol cloudy-sky portion. Table 3 shows that in July 2018, on a pixel basis, the derived $NO_2$ in Case 3 are larger than those in Case REF, with an average difference increasing from 3.1% at relatively clean situations ($NO_2$ VCDs in Case REF $< 5 \times 10^{15}$ molecu· $cm^{-2}$) to 11.2% for polluted situations ($NO_2$ VCDs in Case REF $\geq 15 \times 10^{15}$ molecu· $cm^{-2}$). The spatial distributions in Fig. S1a and S1b also show higher $NO_2$ VCDs in Case 3 relative to Case REF. The corresponding increases in CF (Fig. S1c versus S1d) are because in Case 3 the scattering contributions to the radiance from aerosols in the cloudy-sky portion (that would have occurred)

are accounted for with higher CFs. The enhanced "shielding effect" of clouds (due to higher CFs) result in lower $NO_2$ AMFs and higher VCDs.

For surface reflectance, Case REF considers the BRDF effect instead of Lambertian Equivalent Reflectivity (LER) which is used in deriving FRESCO-S clouds. This leads to inconsistency between when the CP is derived and when it is used. The LER data used by FRESCO-S are generated at 758 and 772 nm (based on the Global Ozone Monitoring Experiment-2), rather than at the 437.5 nm for the $NO_2$ retrieval. Thus Case 4 adopts the OMI LER data from TM5-MP-DOMINO, a five-year monthly based climatology at 440 nm, and re-calculates CFs and $NO_2$ AMFs with explicit aerosol corrections and FRESCO-S CP (Table 2). Here, the ice-snow flag in the TM5-MP-DOMINO product is used to exclude the possible ice/snow contamination, and only the pixels with blue-sky albedos (derived from the BRDF data in Case REF) less than 0.3 are taken into consideration. The resulting $NO_2$ VCDs in Case 4 are lower than Case REF by 3.7% on pixel-based average for relatively clean situations and by 8.3% for polluted situations (Table 3). Figure S2a and S2b further shows the spatial distributions of the relative and absolute differences in derived monthly mean $NO_2$ VCDs between Case 4 and Case REF. In general, Case 4 leads to lower $NO_2$ VCDs than Case REF because of stronger surface reflectance, as is obvious in the comparison of blue-sky albedo in Case REF and LER albedo in Case 4 (Fig. S2c versus S2d). "

**Specific comments:**

Figure 1: Reflectance at 758nm -> 437.5nm.

Changed.

Figures 6-7: these figures can be merged. I suggest to add similar figures where the y-axis use retrieved MAX-DOAS profiles smoothed by satellite averaging kernels. This will remove the error on the satellite retrieval coming from the profile shape uncertainty.

Figure 7 has been merged with Fig. 6.

It is a good suggestion to show the results using retrieved MAX-DOAS profiles smoothed by satellite averaging kernels. However, we do not have MAX-DOAS $NO_2$ profiles at any station.

Page 17 line 9: more much -> much more

Changed.

SI Table S2: Number of measurements within +/- 0.5h -> Number of measurements within +/- 1h, why?

This phenomenon occurs at Xuzhou. It is because our criterion to ensure weak temporal variability of $NO_2$ within the time window; large variability would suggest very local signals that can contaminate the comparison. Indeed, expanding the sampling (temporal) window leads to much stronger variability at Xuzhou, leading to fewer numbers of valid days (7 days are excluded in Table S2).

In Line 185-187, we stated that "To reduce the influence of local events, we exclude all MAX-DOAS data whose standard deviations within the period exceed 20% of their mean values.".

We have added a footnote in Table S2 to explain this issue.

SI reference is missing.

Added.

**Anonymous Referee #2**

General comments:

The reviewer #1 has concerned on the inconsistent treatment of aerosol corrections between the NO2 AMFs calculation and cloud pressure retrieval. In fact, there are many conflicting considerations due to the hybrid POMINO algorithm itself (i.e., using tropospheric NO2 SCDs and cloud pressure from the operational product). For example, the surface BRDF parameters are used in the tropospheric NO2 AMF calculations, while Lambertian Equivalent Reflectance (LER) are assumed during the retrieval of cloud pressure and tropospheric NO2 SCDs. These conflicting considerations may introduce larger error sources on the NO2 retrieval compared to other consistent independent retrieval algorithm. It should be investigated and discussed in the manuscript.

Thank you very much for your suggestions. Inconsistency often occurs in tracer gas retrievals and is extremely difficult to avoid completely. In this revised manuscript, we have added a section (Section 3.3) and additional experiments to test the effect of such issue.

In Line 161-167, we add:"

Two additional cases (Cases 3 and 4) concern the treatment of CP in combination with the choice of aerosols and surface reflectance. Specifically, using the CP data directly from FRESCO-S means that our retrieval algorithm does not perfectly account for the effect of aerosols on clouds. Our retrieval consider the BRDF effects while Lambertian surface is used in deriving the FRESCO-S CP. "

In Line 347-385, we add:"

*3.3 Influences of directly using the CP data from FRESCO-S*

As we take the CP data directly from the FRESCO-S retrieval rather than re-retrieving CP (as done for CF), two main issues arise. First, the FRESCO-S retrieved CP may be affected by aerosols, thus using such CP data in our explicit aerosol corrections (Case REF) may lead to over-correction of aerosol effects. To estimate the effect of such over-correction on retrieved $NO_2$ VCDs, we employ in an additional sensitivity case (Case 3 in Table 2) a "semi-explicit" aerosol correction approach. This approach explicitly includes aerosols in the calculation of AMF for the clear-sky portion ($M_{clr}$) of a pixel (as in Case REF), but excludes aerosols for the cloudy-sky portion ($M_{cld}$)

of that pixel. Correspondingly, CF is re-calculated on the basis that the radiance at 437.5 nm received by TROPOMI is contributed from the aerosol-contained clear-sky portion and the no-aerosol cloudy-sky portion. Table 3 shows that in July 2018, on a pixel basis, the derived $NO_2$ in Case 3 are larger than those in Case REF, with an average difference increasing from 3.1% at relatively clean situations ($NO_2$ VCDs in Case REF $< 5 \times 10^{15}$ molecu· $cm^{-2}$) to 11.2% for polluted situations ($NO_2$ VCDs in Case REF $\geq 15 \times 10^{15}$ molecu· $cm^{-2}$). The spatial distributions in Fig. S1a and S1b also show higher $NO_2$ VCDs in Case 3 relative to Case REF. The corresponding increases in CF (Fig. S1c versus S1d) are because in Case 3 the scattering contributions to the radiance from aerosols in the cloudy-sky portion (that would have occurred) are accounted for with higher CFs. The enhanced "shielding effect" of clouds (due to higher CFs) result in lower $NO_2$ AMFs and higher VCDs.

For surface reflectance, Case REF considers the BRDF effect instead of Lambertian Equivalent Reflectivity (LER) which is used in deriving FRESCO-S clouds. This leads to inconsistency between when the CP is derived and when it is used. The LER data used by FRESCO-S are generated at 758 and 772 nm (based on the Global Ozone Monitoring Experiment-2), rather than at the 437.5 nm for the $NO_2$ retrieval. Thus Case 4 adopts the OMI LER data from TM5-MP-DOMINO, a five-year monthly based climatology at 440 nm, and re-calculates CFs and $NO_2$ AMFs with explicit aerosol corrections and FRESCO-S CP (Table 2). Here, the ice-snow flag in the TM5-MP-DOMINO product is used to exclude the possible ice/snow contamination, and only the pixels with blue-sky albedos (derived from the BRDF data in Case REF) less than 0.3 are taken into consideration. The resulting $NO_2$ VCDs in Case 4 are lower than Case REF by 3.7% on pixel-based average for relatively clean situations and by 8.3% for polluted situations (Table 3). Figure S2a and S2b further shows the spatial distributions of the relative and absolute differences in derived monthly mean $NO_2$ VCDs between Case 4 and Case REF. In general, Case 4 leads to lower $NO_2$ VCDs than Case REF because of stronger surface reflectance, as is obvious in the comparison of blue-sky albedo in Case REF and LER albedo in Case 4 (Fig. S2c versus S2d). "

The manuscript lacks of necessary uncertainty estimation on tropospheric $NO_2$ retrieval, considering the existing conflicting considerations in the hybrid retrieval algorithm. Also, a reasonable error budget is beneficial to the potential users.

In addition to discussing the issue related to FRESCO CP, we have added a new Section 3.4 (Line 391-407) to discuss the errors related to individual parameters: "

3.4 Error estimates for POMINO-TROPOMI $NO_2$ AMFs

It is difficult to derive an overall AMF error for each pixel with our algorithm, particularly because of the three-dimensional aerosol parameters used, the inter-linkage between clouds and other parameters (aerosols and surface reflectance), and the exclusion of LUTs (which leads to too computationally expensive error estimates). Table 4 provides a preliminary estimate of the $NO_2$ AMF errors with respect to uncertainties in individual parameters. We follow the ATBD of TM5-MP-DOMINO (van Geffen et al., 2019) and make use of error estimates in previous studies. Individual errors with respect to CF, CP and BRDF coefficients are within 10%. Errors with respect to aerosols are considered to be larger, due to uncertainty in AOD, SSA and aerosol vertical profiles (Liu et al., 2019), as well as the fact that using the CP from FRESCO-S rather than deriving it here may lead to an additional error in the $NO_2$ AMFs. Note that these individual errors are not

fully independent due to the aforementioned linkage between parameters. The magnitude of potential systematic bias in $NO_2$ may be smaller than the quadrature sum of errors in individual parameters, as suggested by the slight mean bias relative to MAX-DOAS data (Fig. 6a)."

In Fig. 6, only limited MAX-DOAS observations on Summer 2018 (N=63) are used to validate satellite NO2 retrieval. However, aerosol consideration can make larger difference to NO2 retrieval especially in winter with severe haze pollution. It would be more convincing to include more data in temporal coverage.

We agree that more data for comparison will be more convincing. However, we have tried our best to collect the MAX-DOAS dataset. At a later stage, we hope to collect more data in different seasons to extend the comparison.

Nevertheless, our results show that the KNMI official product largely underestimates MAX-DOAS measurements, which is consistent with previous findings (Griffin et al., 2019; van Geffen et al., 2019). This consistency gives additional confidence in drawing our conclusion here. Besides, since our TROPOMI algorithm is quite similar to that for OMI (i.e., POMINO v2), and our previous evaluation for POMINO v2 shows that even in very polluted (wintertime) cases, our algorithm captures the magnitude and variability in MAX-DOAS data very well, and is better than QA4ECV (Liu et al., (2019).

Specific comments:

Page 4, Line 8-9: compared to the previous POMINO version in Liu et al., 2019, constraining aerosol profile with CALIPSO data is no longer used in this paper. Why? It would be better to explain these algorithm changes.

We have chosen to add CALIOP aerosol profiles in our TROPOMI algorithm at a later stage, when we will be able to use the $O_2$-$O_2$ cloud algorithm to re-calculate both CF and CP.

We have added a sentence in Line 145-147:"Our future study will use the CALIOP aerosol extinction vertical profiles to further improve the modeled aerosol profiles."